# Low profile high gain RHCP antenna for L-Band and S-Band using rectangular ring metasurface with backlobe suppression

**Sundas Farooq Khan, Bilal Muhammad Khan** *, **Tariq Mairaj Rasool Khan**

National University of Sciences and Technology, Karachi, Pakistan

* bmkhan27@gmail.com

**Data Availability Statement:** All relevant data are within the manuscript.

**Funding:** The author(s) received no specific funding for this work.

## Abstract

In this reported work a single feed, miniaturized, dual layer, and low profile antenna is presented for 1.575GHz frequency band. The proposed antenna offers high gain, lower noise bandwidth, with better sensitivity and range. The ground choke technique is used for back lobe suppression. The prototype is fabricated on FR 4 substrate using manual fabrication technique. This offers an inexpensive and readily available fabrication. Therefore, fabricated antenna is small size, low cost, easily fabricated and tested for satellite communication. The antenna comprises of two layers, containing a patch radiator and a Metasurface layer with 3x3 rectangular ring resonators. The layers are separated using foam with a 12mm width. The proposed prototype is radiating at 1.575GHz and 2.33GHz with an overall dimension of 85.6 x 68.4 x 15.204 mm. The developed antenna provides a gain of 5.9 dBi. The simulated results are verified using VNA and anechoic chamber testing. Moreover, the developed antenna has been successfully tested for L-Band Satellite communication in real time scenario without any LNA. Higher Gain due to Metasurface increase the efficiency of the system. The promising results indicate the aptness of the developed antenna for real-world applications of L-Band and S-Band.

## 1. Intorduction

METASURFACES are two-dimensional metamaterials i.e. a geometry spatially arranged in such a way that it exhibits unusual but homogenous properties so it can be used as a single building block and facilitates general application. According to Snells's law, altering surface impedance create phase shift so by altering impedance of the surface we can control the reflection parameters. Metasurface consist of arrays of subwavelength scatterers that can be designed to manipulate the phase, amplitude, and polarization of electromagnetic waves in a highly controlled manner. By introducing variable phase discontinuities across the surface, desired reflection and refraction properties can be achieved. Thus, making it suitable for variety of applications. They have been widely studied in the field of optics and have been used to create a wide range of functional devices such as waveplates [1–3], polarizers [4–8], and beam shapers [9–13]. One of the key advantages of metasurfaces is that they can be used to create highly compact and

**Competing interests:** The authors have declared that no competing interests exist

lightweight devices that can be integrated into a wide range of systems. Extensive research has been made in the field of metasurfaces with expensive and customized substrates with multi-layer design and complex printing procedure. There is a dire need to explore the field with commonly available materials and simple configuration.

The growing use of satellite communication has made it a crucial tool for connecting the disconnected [14]. The goal of satellite communication is reliability. Although higher frequency band give higher bandwidth and smaller antenna size however they are more susceptible to signal degradation also known as rain fading. Therefore, lower frequency bands are preferred when reliability is required. Using metasurface over conventional antenna not only reduce size but also increase gain and bandwidth without compromising on reliability of the system. Moreover, circular polarized antenna is preferred as they are insensitive to the physical alignment of receiver. High gain at narrow bandwidth reduces the noise bandwidth at receiver end hence provide better sensitivity and higher read range. A polarization converter is presented in [15] for frequency 4.386.32GHz with 6.05 dBic gain. A reconfigurable antenna with 5.6dBi gain for 5-6GHz frequency range is presented in [16] for multidirectional beam formation. In [17] S-shaped Metasurface antenna is designed for 5.3–6.6 GHz. An annular ring slot design report results for 4.8–7.5GHz range in [18]. In [19, 20] low profile antennas are presented for higher frequency spectrum. However, [21–23] are presented for higher frequency ranges than 1.575GHz of GPS band.

Survey on Metasurface literature suggests very few papers have been published for L-Band. Furthermore, no low profile (FR 4 based) Metasurface antenna is reported for L band to the best of author's knowledge. In summary, this work introduces a novel low-profile metasurface antenna designed for the L-band in satellite communication. Utilizing commonly available materials and a simplified configuration, the proposed antenna aims to reduce size, enhance gain and bandwidth, while maintaining system reliability. Addressing a notable gap in existing literature, this research contributes to the exploration of metasurface applications in satellite communication, particularly in the less-explored L-band frequency range using commonly available FR 4 substrate

## 2. Design and fabrication

The proposed design consists of two layers. First layer consists of a patch antenna with coaxial feed. This layer has ground plan with etched slots in it for back lobe suppression. Second layer consist of Metasurface unit cells. This layer does not have direct feeding or ground plan.

Patch Antenna is designed using governing design equations of microstrip patch antenna [24]. Length of the patch was calculated with (1). Where f is resonating frequency and $\varepsilon_r$ permittivity of substrate is 4.3. From (1) calculated length of patch is 45mm. However, best results were observed with rectangular patch with aspect ratio of 1.15 therefore, length and width is 48.5mm and 42.175mm respectively.

$$L = \frac{V_o}{2f\sqrt{\varepsilon_r}} \tag{1}$$

whereas feed point was calculated by locating point where input impedance matches 50Ω

$$x_o = x_f \times \cos(\theta) \tag{2}$$

$$y_o = x_f \times \sin(\theta) \tag{3}$$

Where $x_f$ = 17.0 and $\theta$ = 35 degree while feed is in the 2nd quadrant. Backward radiation is generated at patch antenna on the finite ground plane due to the ground plane edge diffraction

**Table 1. Dimensions of parameter.**

| Parameter | W | L | W1 | L1 | W2 |
|---|---|---|---|---|---|
| Value (mm) | 85.6 | 68.4 | 48.5 | 42.17 | 20 |
| Parameter | L2 | W3 | L3 | W4 | L4 |
| Value (mm) | 17.4 | 4 | 4 | 5.8 | 30 |

[25]. The slotted ground choke is created by etching four slots at the corner of the ground plane of the FR4 substrate as dimensions tabulated in Table 1.

Rectangular rings are subwavelength scatterer which are $\approx \lambda/10$ with aspect ratio of 1.14. It is studied in [26] that rectangular scatterers give wider bandwidth. Aspect ratio is a characteristic of substrate.

A complete model of antenna is presented in Fig 1. Parameters are tabulated in Table 1.

Functionality of Metasurface layer can be understood through superstrate or resonant cavity Antenna, Metasurface layer can be applied as superstrate or around patch to suppress surface wave or reshape [27–29]. By appropriately designing the scaterrers and controlling the phase and amplitude of wave it's possible to enhance the amplitude of the waves in the desired direction, effectively increasing the gain of the antenna in that direction. Left- handed

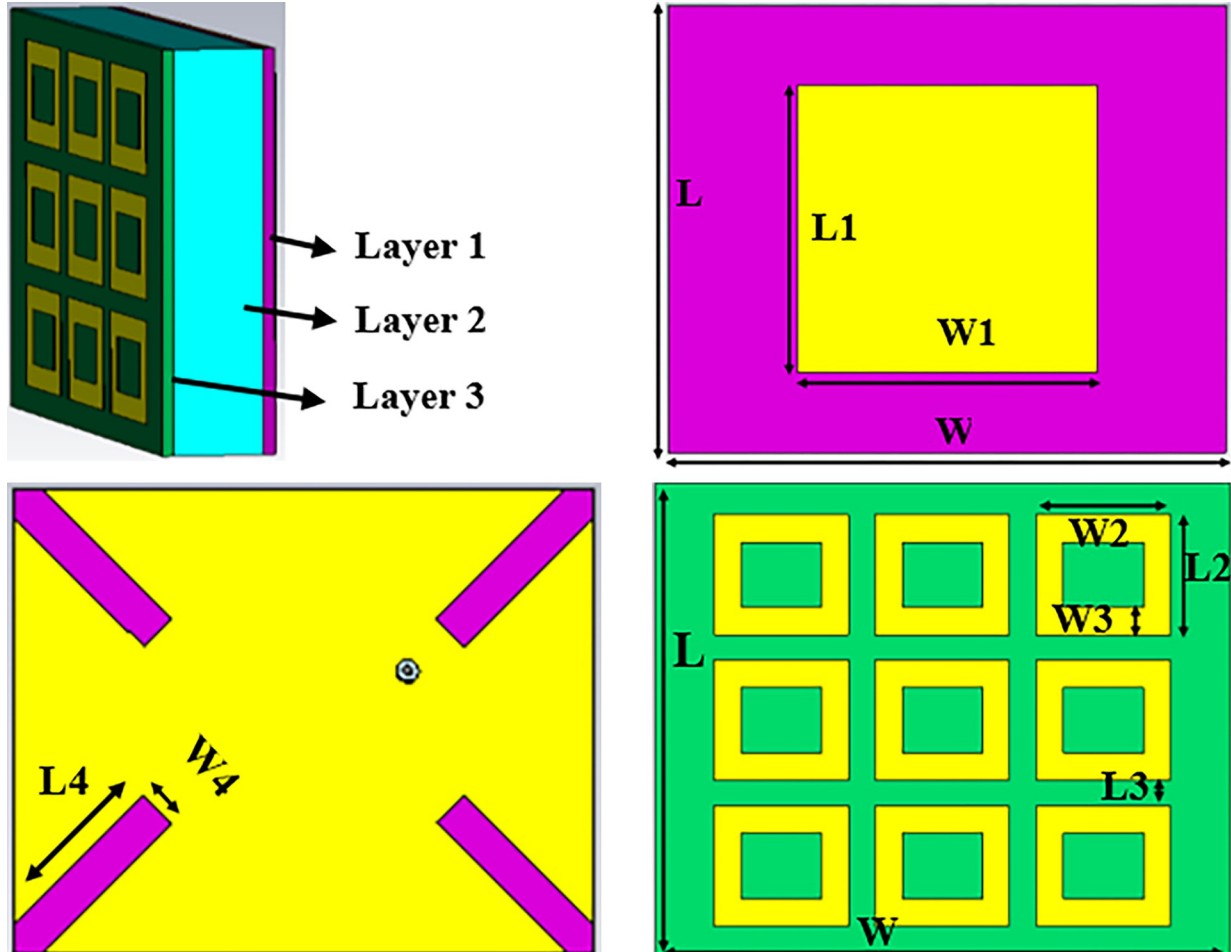

**Fig 1.** Antenna Configuration: (a)Perspective view, (b) Coaxial fed rectangular patch antenna (c) Ground plan (d) Metasurface Layer.

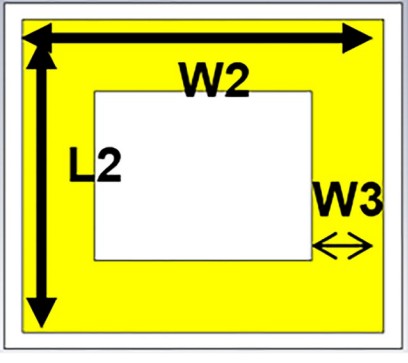 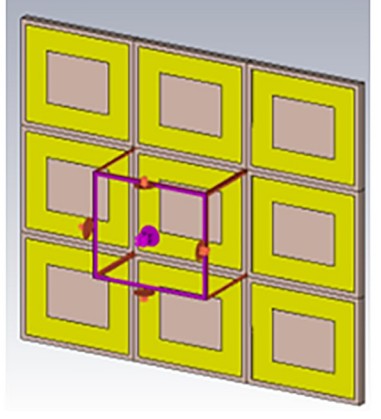 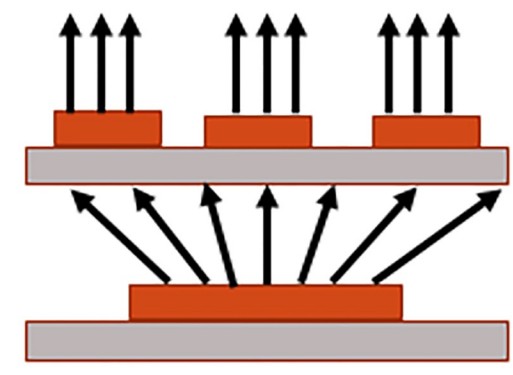

**Fig 2.** (a)Unit Cell (b)Floquet Port View in CST (c) Working Principle Design & Methadology.

Metametrial (LH-MTMs) or Double negative (DNG) materials as well as Electronic bandgap structure (EBG) also enhance gain and directivity when primary source antenna is paired with them [30–32]. In [27] double closed ring resonator (DCR) as reflective metasurface unit cell was investigated. It was found that at 2.2 GHz and 2.9GHz DCR shows unusual properties such as permeability close to zero. Retrieved effective parameters shows DCR lies in Mu Near Zero (MNZ) & Epsilon Negative medium (ENG) region at 2.2 and 2.9GHz frequency thus enhancing gain.

For our desired frequency i.e. 1.575GHz we designed the single closed ring resonator with bigger dimensions as shown in Fig 2(A). Only unit cell was also designed separately through floquet port of CST MW. It was concluded that metasurface in subject case is collimating the wave nature thus increasing directivity and gain. Further study on retrieved effective parameters is in progress.

Manual fabrication technique is used for prototype development. This includes etching of substrate through Ferric Chloride solution. A fabricated prototype is presented in Fig 3.

## 3. Meaurement & testing

Antenna was tested on vector network analyzer for reflection parameter. It was then tested in an anechoic chamber for gain and radiation patterns. Fig 4. Shows the VNA testing and anechoic chamber testing.

Developed Antenna is tested for satellite communication using test bench. Open ground testing was conducted to receive real GPS signals. Prototype was connected with USRP X310

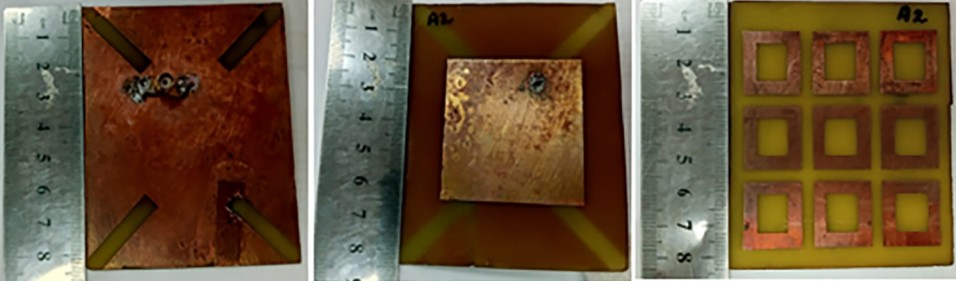

**Fig 3. Fabricated antenna.**

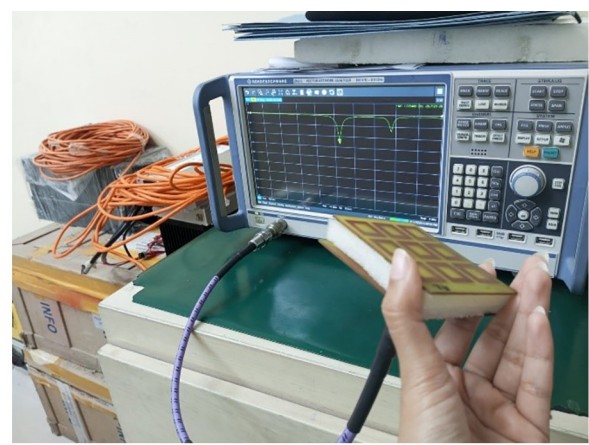

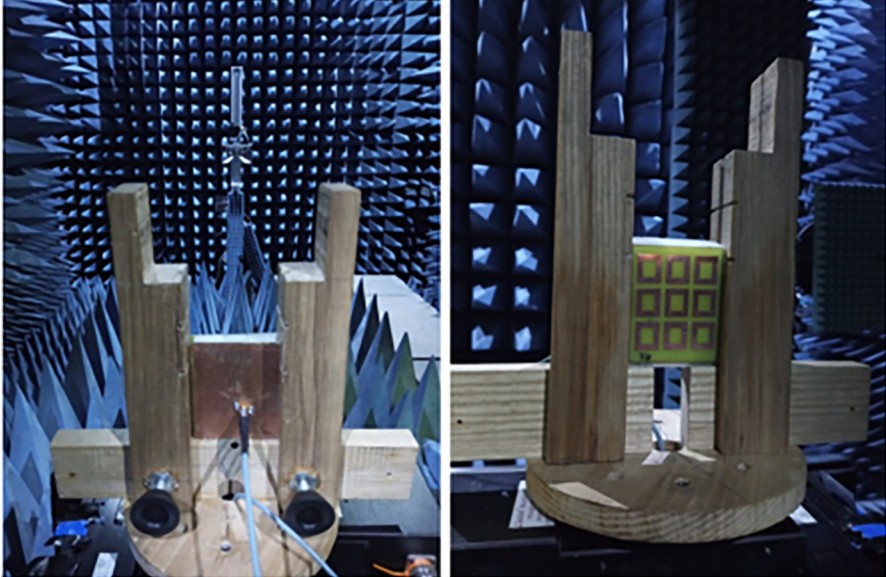

**Fig 4.** Testing of Prototype (a) VNA testing (b) Anechoic Chamber testing.

SDR without any external LNA. Test setup is shown in Fig 5(A). Whereas Fig 5(B). explain the block diagram of test setup. Device under test (DUT) is connected with Software Defined radio (SDR) An SDR is a transceiver system consisting of a radio front end (RFE) and a digital back end with a variety of on-board DSP capabilities. This test involved analyzing the received signal for visualizing how many satellites were captured also utilizing a GNU Radio flowgraph to measure the power of the signals received by DUT.

## 4. Result and discussion

The developed Antenna is designed for satellite communication and is functional for 1.575GHz (L-band) as well as for 2.33GHz (S-Band) with reflection coefficient less than -10dB. Fig 6. Shows measured and simulated results. In order to qualify antenna for satellite communication reception its reflection coefficients, polarization and gain was checked. Simulated results show less prominent reflection parameter at 2.33 GHz. However, measured results show that antenna is capable of transmitting 90% of the power at 2.33GHz as well as, S11 is below -10dB at said frequency. Fig 7. Shows axial ratio of antenna which is below for entire

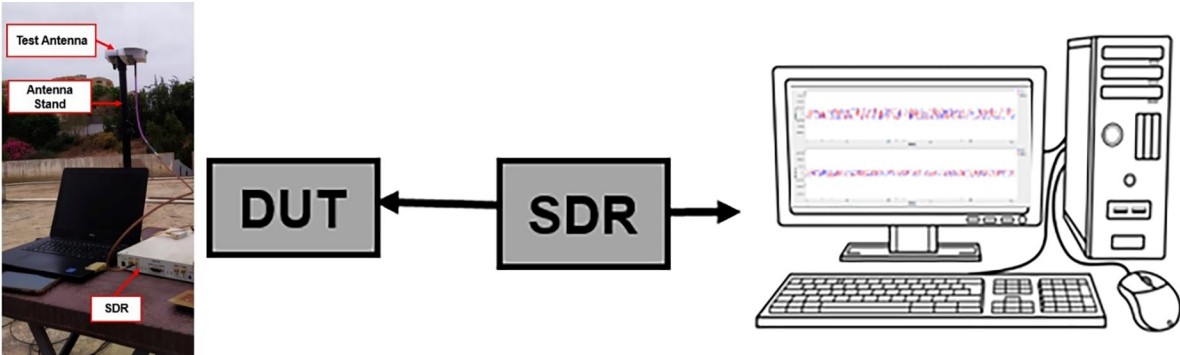

**Fig 5.** Open ground Testing of Prototype (a) Test Setup (b) Block Diagram of Test Set Up.

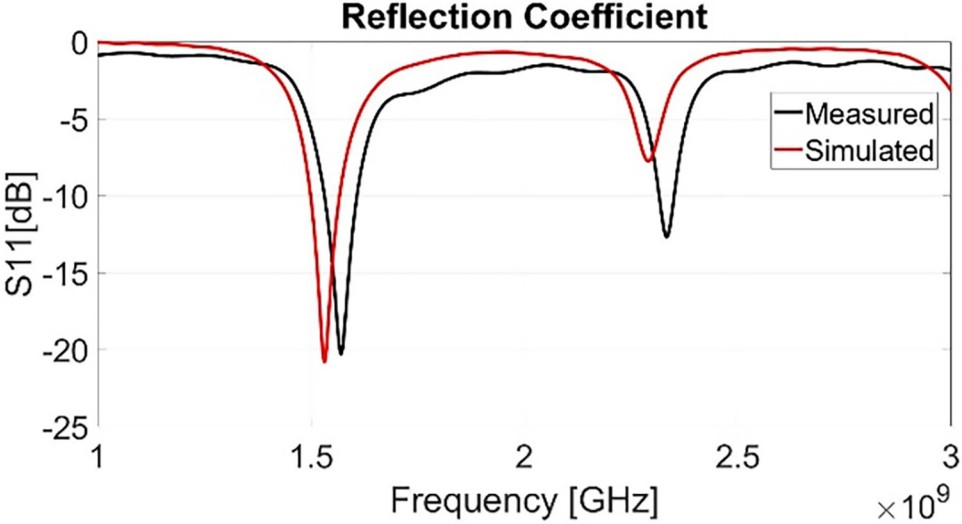

**Fig 6. Measured and simulated S11 of antenna.**

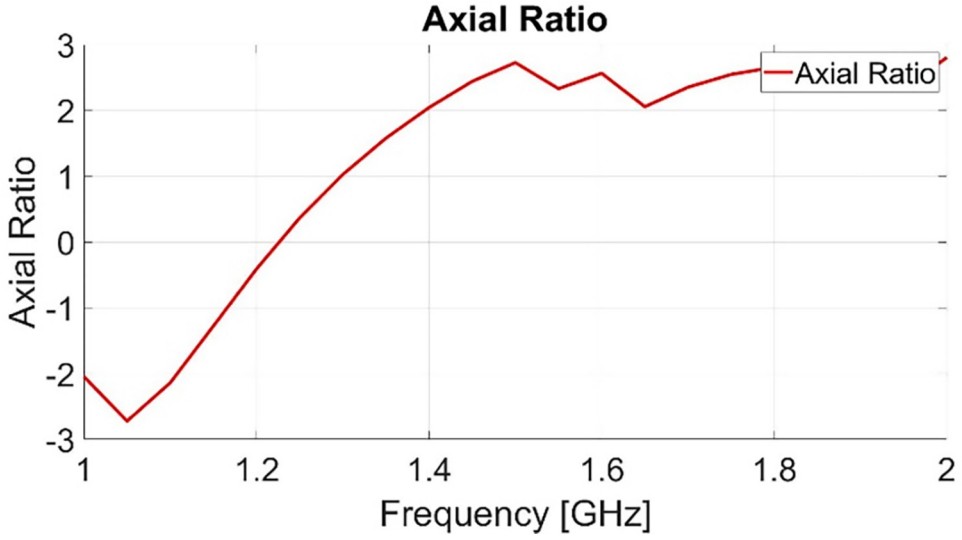

**Fig 7. Axial ratio of antenna.**

Farfield Directivity Abs (Phi=90)

— farfield (f=1.575) [1]

Phi= 90    Phi=270

Theta / Degree vs. dBi

Frequency = 1.575 GHz
Main lobe magnitude =    5.91 dBi
Main lobe direction =    1.0 deg.
Angular width (3 dB) =   104.8 deg.
Side lobe level =  -12.1 dB

Farfield Directivity Left Polarisation (Phi=90)

— farfield (f=1.575) [1]

Phi= 90    Phi=270

Theta / Degree vs. dBi

Frequency = 1.575 GHz
Main lobe magnitude =    -1.22 dBi
Main lobe direction =    3.0 deg.
Angular width (3 dB) =   118.2 deg.
Side lobe level =    -5.0 dB

Farfield Directivity Right Polarisation (Phi=90)

— farfield (f=1.575) [1]

Phi= 90    Phi=270

Theta / Degree vs. dBi

Frequency = 1.575 GHz
Main lobe magnitude =    4.98 dBi
Main lobe direction =    1.0 deg.
Angular width (3 dB) =   102.3 deg.
Side lobe level =  -27.7 dB

**Fig 8.** Radiation pattern after Backlobe Suppression (a)Absolute Gain 5.91 dBi (b) LHCP Gain -1.22dBi (c) RHCP Gain 4.976dBi.

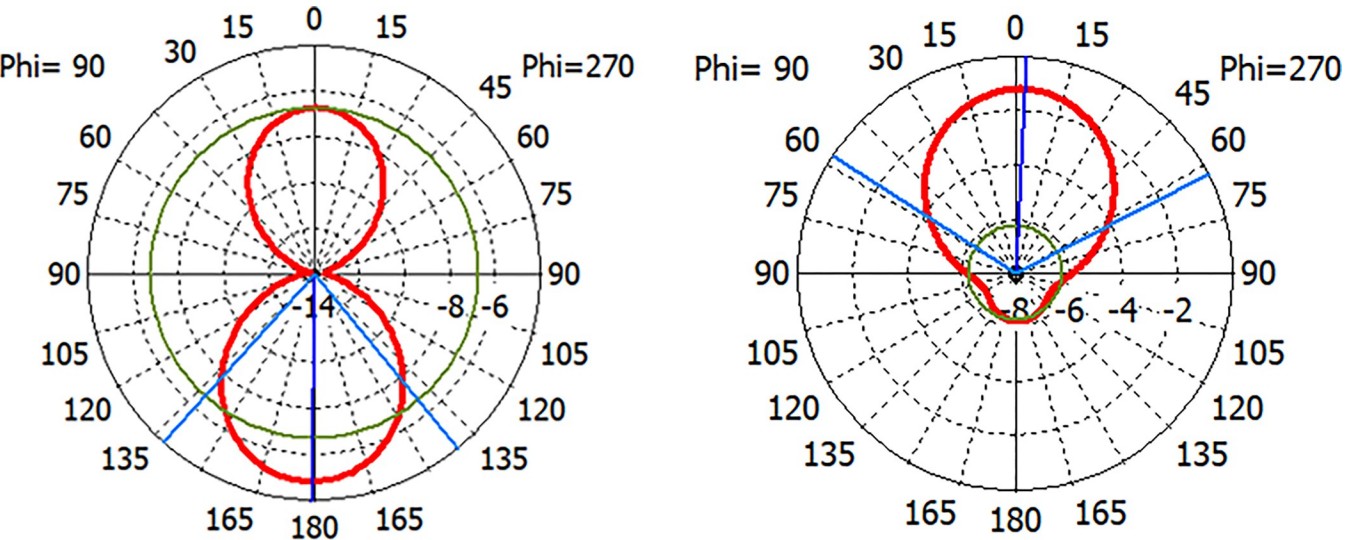

**Fig 9.** Radiation pattern Before & after Backlobe Suppression (a) Without Choke (b) after Choke.

spectrum of 1.575GHz thus circularly polarized. From Fig 6 it is evident that bandwidth is quite narrow at both frequencies thus providing low noise figure. It was learned when a COTS (commercial of the shelf) LNA with 5dB noise figure was used as external amplifier. This specific COTS LNA was chosen due its low cost. In this experiment no satellite communication was established due high noise figure on channel. Each component on channel adds noise on received signal. It is concluded through experiments that noise is an important factor to take into account while reception. Therefore, having narrow bandwidth filters out the unwanted signals at desired frequency thus better.

Antenna is circularly polarized at 1.575GHz however it is linearly polarized at 2.33GHz. But it can be further optimized to make it circularly polarized at 2.33GHz as well. Radiation pattern of antenna is presented in Fig 8. Antenna offers only -1.22dBi Gain in LHCP and 4.98dBi Gain in RHCP regions making it a right hand circularly polarized antenna. Overall Gain of the antenna is 5.91dBi. A 104 degree beam width ensures excellent hemispherical coverage.

Back lobe suppression technique reduced the back lobe. Before fabricating choke on ground plane antenna had considerable back lobe. A side-by-side comparison of impact of choke is presented in Fig 9. It is evident from graph, that without Choke more power was transmitted backward than in beamforming region. This not only reduce the efficiency of antenna but also could cause a potential hazard of unwanted RF exposure.

Measured results of antenna in anechoic Chamber is presented in Fig 10.Testing was done using a linearly polarized horn antenna at transmitting end and rotating it. Fig 10A) shows the radiation pattern in polar plot while Fig 10(B) shows normalized values in Cartesian plot. It can be seen in normalized values that difference of power between co polarized and cross polarized at main lobe is very less prove it to be circularly polarized.

The working of the antenna can be explained with the behavior of wave. A guided wave is a plane wave while the free-space wave is a spherically expanding wave. An antenna is region of transition between guided wave to free space. The presented technique breaks the region of transition into two layers, first layer i.e. patch antenna, radiate. Whereas the second layer collimates the spherically expanding wave with sub wavelength scatterer. This second layer is placed at an optimized distance to have constructive interference.

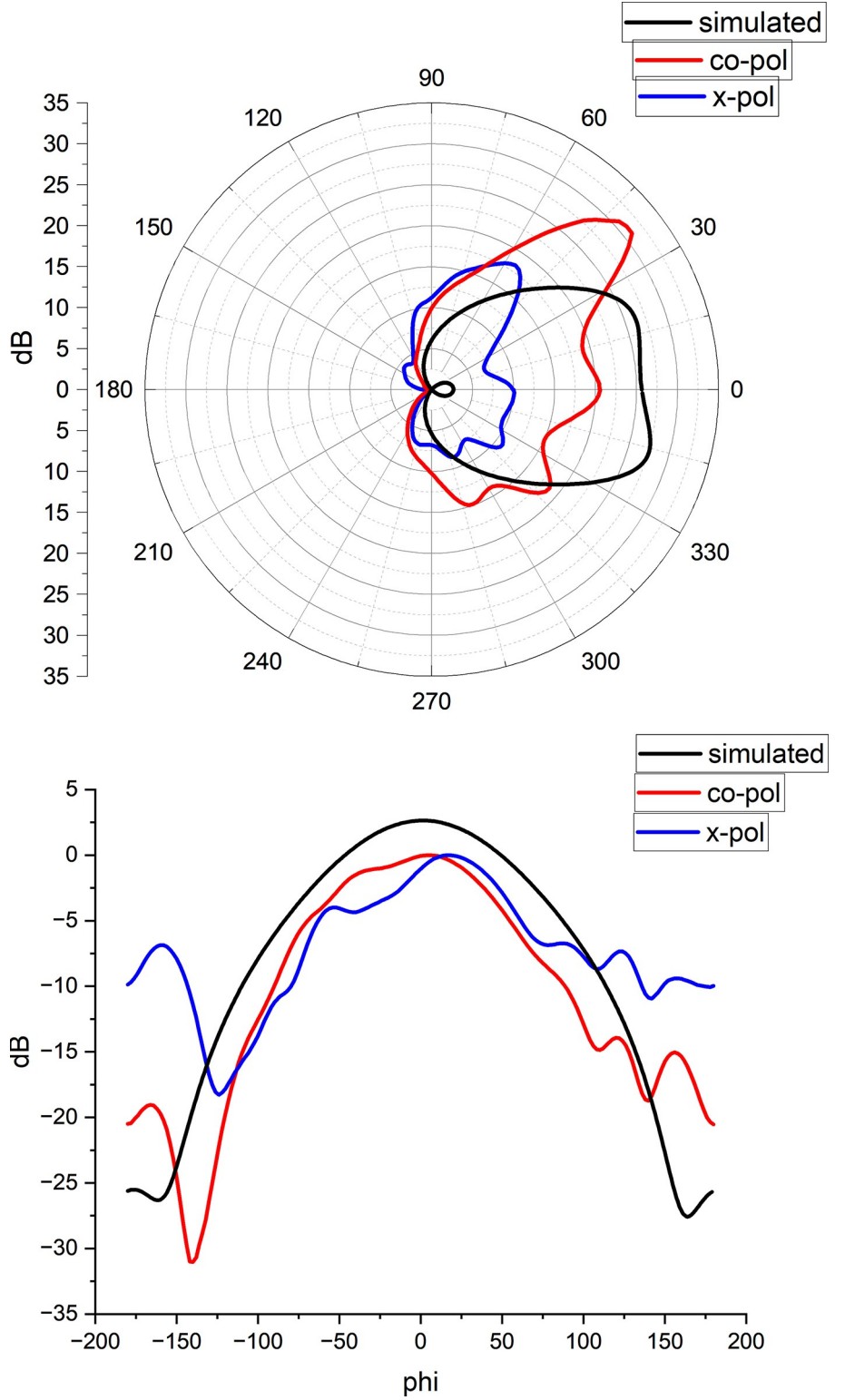

**Fig 10.** Measured Radiation pattern of antenna (a) Polar Plots (b) Normalized Plots.

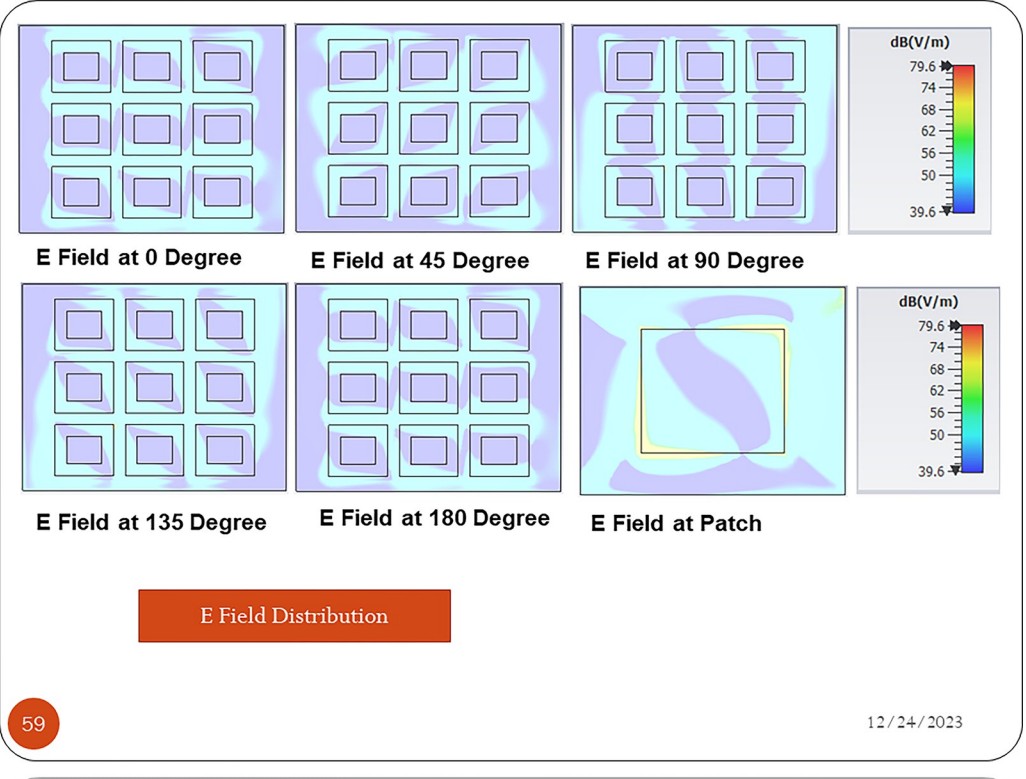

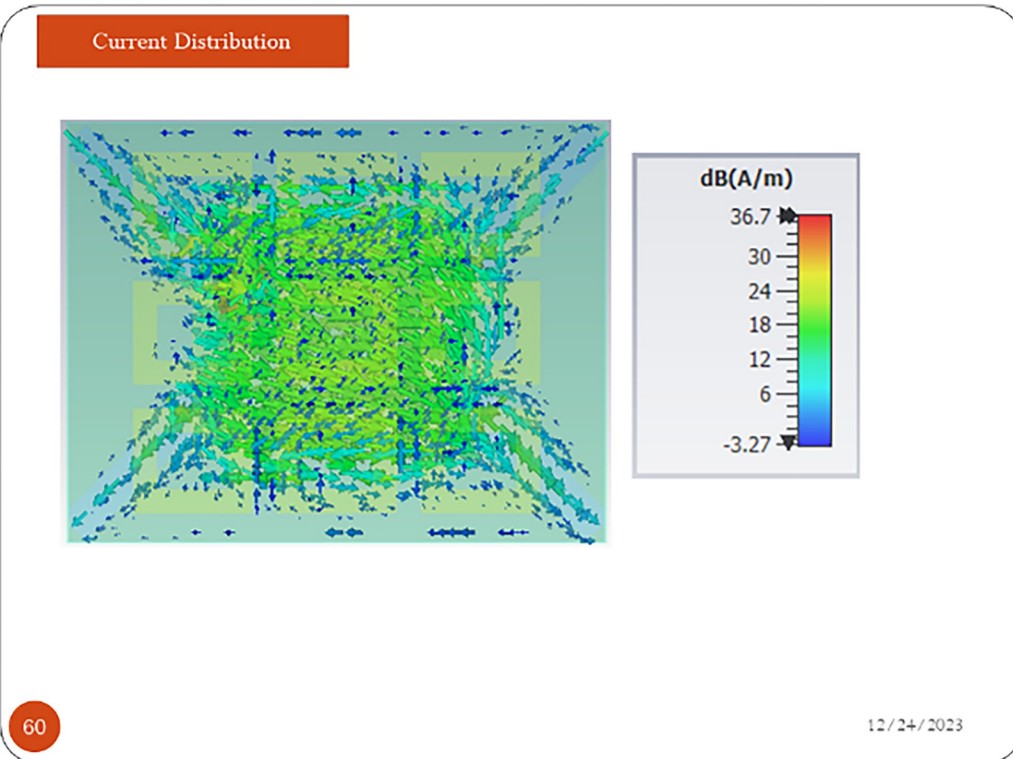

**Fig 11.** (a) Electric Field Scattering at 1.575GHz (b) Surface Current Distribution.

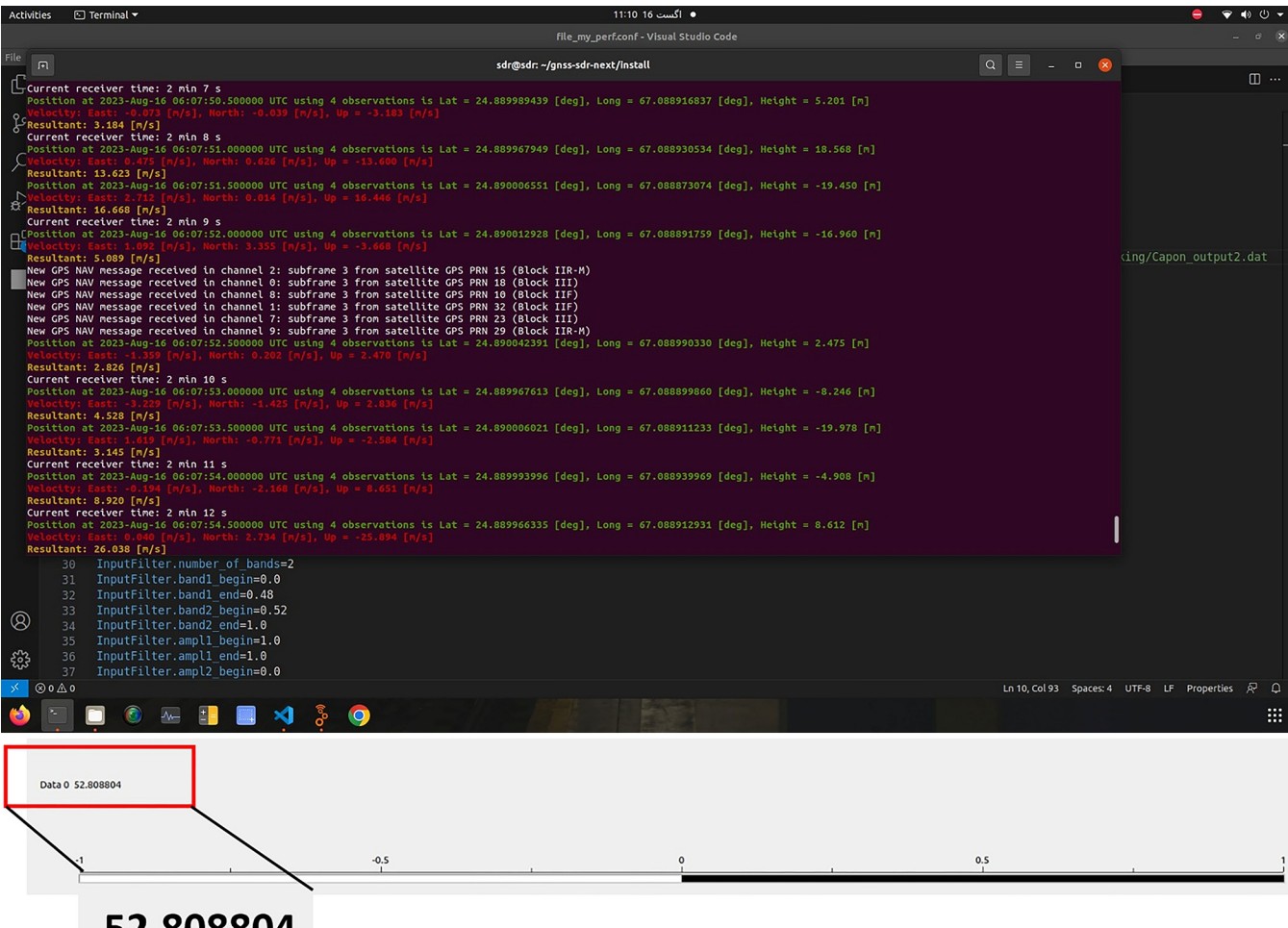

**Fig 12.** (a) 06 Satellites Received (b) Power Received at Flowgraph.

The Metasurface layer or the 2nd layer not only improves gain but also reshapes the reflection coefficient parameter by shifting it to its left. With Metasurface layer, S11 of antenna improves impedance matching as low as below -20dB.

Fig 11(A) presents the Electric field scattering at 1.575GHz and different time variants. It can be seen from the pattern that electric field in patch as well as in each Metacell is rotating in circular motion. Surface current distribution presented in Fig 11(B) shows majority of current is in center of patch and Metasurface. While choke at ground is bringing edge current back to center thus suppressing backlobe.

Test setup successfully track and locked the position of 06 satellites present in the region as shown in Fig 12(A). Once channel is established position velocity and time of the Satellite was calculated. Once satellite started sending subframes to the receiver channel is successfully established. Antenna successfully established links with six satellites. This data acquisition and tracking validated the hypothesis that it can be used for satellite communication. Received Gain at Flowgraph was 52.80dB as presented in Fig 12(B).

The presented prototype has an edge over its predecessor in size reduction, gain enhancement and simplicity. As previously reported, work for same frequency i.e., 1.575GHz was fabricated using customized Rogers substrate. In [33] 2-layer design was presented with 9dBi gain

**Table 2. Comparison with other paper.**

|  | Proposed Design | Paper [33] | Paper [34] |
|---|---|---|---|
| Frequency | 1.575GHz | 1.575GHz | 1.575GHz |
| Size (mm) | 85.6x68.4 x15.204 | 175x175x21 | 75x75x13 |
| Gain | 5.9dBi | 9dBi | 5dBi |
| Substrate | FR 4 | Rogers Customized | Rogers Customized |
| Layers | 2 | 2 | 3 |

but with large dimensions. In [33] 3-layer design was presented adding complexity in manufacturing process and 5dBi gain. This paper presents a design which gives more than 50% size reduction over [33] and better gain than [34]. Moreover, it has added advantage of easy fabrication on readily available substrate. A comparison is delineated in Table 2.

## 5. Conclusion

A novel rectangular ring shaped Metasurface based patch antenna with slotted ground choke is presented for L-band and S-band. Antenna is circularly polarized due to its diagonal feed. Stacked approach of antenna enhanced the overall Gain of antenna. It is fabricated on FR 4 substrate and can be used for satellite communication. The developed antenna, with an overall dimension of 85.6 x 68.4 x 15.204 mm provides a gain of 5.9 dBi. The simulated results are verified using VNA and anechoic chamber testing. Moreover, the developed antenna has been successfully tested for L-Band Satellite communication for data acquisition of commercial satellites. The promising results indicate the efficacy of the developed antenna for real-world applications of L-Band. Thus fabricated prototype antenna is small size, low cost, easily fabricated and readily available for satellite communication. Study of retrieved effective parameters, further optimization of antenna for S-Band to increase multi-functionality is the proposed future work. Further size reduction and exploring the design on different substrate are also termed as future activities.

## Author Contributions

**Conceptualization:** Sundas Farooq Khan, Bilal Muhammad Khan.

**Data curation:** Sundas Farooq Khan, Tariq Mairaj Rasool Khan.

**Formal analysis:** Sundas Farooq Khan, Bilal Muhammad Khan.

**Methodology:** Sundas Farooq Khan, Bilal Muhammad Khan, Tariq Mairaj Rasool Khan.

**Project administration:** Bilal Muhammad Khan.

**Software:** Tariq Mairaj Rasool Khan.

**Supervision:** Bilal Muhammad Khan.

**Writing – original draft:** Sundas Farooq Khan.

**Writing – review & editing:** Sundas Farooq Khan, Bilal Muhammad Khan.

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
