## [Decision Letter · Decision Letter 0]

14 Nov 2023

PONE-D-23-33755Low Profile High Gain RHCP Antenna for L- Band and S-Band using Rectangular Ring Metasurface with Backlobe suppressionPLOS ONE

Dear Dr. Khan,

Thank you for submitting your manuscript to PLOS ONE. After careful consideration, we feel that it has merit but does not fully meet PLOS ONE’s publication criteria as it currently stands. Therefore, we invite you to submit a revised version of the manuscript that addresses the points raised during the review process.

We look forward to receiving your revised manuscript.

Kind regards,

Yuan-Fong Chou Chau

Academic Editor

PLOS ONE

Journal Requirements:

4. Thank you for stating the following in your Competing Interests section: "The Authors have no competing interest to report"

Reviewers' comments:

Reviewer's Responses to Questions

**Comments to the Author**

1. Is the manuscript technically sound, and do the data support the conclusions?

Reviewer #1: Yes

Reviewer #2: Yes

2. Has the statistical analysis been performed appropriately and rigorously? 

Reviewer #1: Yes

Reviewer #2: N/A

3. Have the authors made all data underlying the findings in their manuscript fully available?

Reviewer #1: Yes

Reviewer #2: Yes

4. Is the manuscript presented in an intelligible fashion and written in standard English?

Reviewer #1: Yes

Reviewer #2: Yes

5. Review Comments to the Author

Reviewer #1: Dear authors;

1-Can you summarize in a few sentences what all this work is about at the end of the introduction?

2-Theabstract needs more interest and rewriting of some paragraphs.

3-There are still some aspects that can be improved (for grammar and punctuation). Improve the technical writing of your paper, where there are several grammatical errors and spelling I think they need to be checked out.

4-The conclusion needs more effort to elaborate on the achieved results with respect to future work,

5-There are still some aspects regarding the obtained results discussions that are missing. Can you please address your achievements well?

6-The practical part is very important, therefore, I’m asking about the radiation pattern measurements,

7-Why this design geometry, explain the design methodology and specify the novelty of your antenna structure,

8-Future work is an important part of the conclusion.

9-The results are still not matured well. Can you clearly mention what your objectives clearly for each result you got during the discussion process?

10- There are several relative works, I wish to discuss, based on adding such structures to the antenna. Can you please suggest your enhancements and advancements (if you would) over such as the following:

A- https://doi.org/10.1016/j.aeue.2023.154652

B- https://doi.org/10.1002/mop.33761

C- https://doi.org/10.36244/ICJ.2023.2.1

D- https://doi.org/10.32620/reks.2023.1.06

E- https://doi.org/10.1002/mop.33666

I loved this work and I feel it is very good. I hope these comments will help you improve this work after a major revision.

Regards

Reviewer #2: The authors presented their work, "Low Profile High Gain RHCP Antenna for L- Band and S-Band using Rectangular Ring

Metasurface with Backlobe suppression", which seems to be quite interesting and important from a research point of view considering its outcomes. However, some technical modifications are required before considering revising it.

First of all, the authors are required to highlight the need for the proposed metasurface antenna design by explaining its pros and cons over the ones reported in the existing literature. So, a brief inclusion of it is required in the manuscript.

The analysis of the metasurface unit cells is missing. On what basis it is working in the proposed antenna design needs to be explained.

Considering the attainment of circular polarization (CP) outcomes, it is significant to add the analysis in terms of electric field distribution, surface current distribution, radiation pattern, etc. In the radiation pattern, LHCP and RHCP components should be shown clearly.

The authors should try to plot the obtained results by using the plotting software.

Finally, a comparison table must be prepared, highlighting the performance aspects, mainly CP characteristics, of the proposed metasurface antenna compared to others.

6. PLOS authors have the option to publish the peer review history of their article (what does this mean?). If published, this will include your full peer review and any attached files.

Reviewer #1: No

Reviewer #2: **Yes: **Dr. Bikash Ranjan Behera

---

## [Author Response · Author response to Decision Letter 0]

29 Dec 2023

All reviewer and Editorial Comments are addressed . Please find attach Response to Reviewer file for further Information.

---

## [Decision Letter · Decision Letter 1]

16 Jan 2024

Low Profile High Gain RHCP Antenna for L- Band and S-Band using Rectangular Ring Metasurface with Backlobe suppression

PONE-D-23-33755R1

Dear Dr. Khan,

We’re pleased to inform you that your manuscript has been judged scientifically suitable for publication and will be formally accepted for publication once it meets all outstanding technical requirements.

Kind regards,

Yuan-Fong Chou Chau

Academic Editor

PLOS ONE

Additional Editor Comments (optional):

Reviewers' comments:

Reviewer's Responses to Questions

**Comments to the Author**

1. If the authors have adequately addressed your comments raised in a previous round of review and you feel that this manuscript is now acceptable for publication, you may indicate that here to bypass the “Comments to the Author” section, enter your conflict of interest statement in the “Confidential to Editor” section, and submit your "Accept" recommendation.

Reviewer #1: All comments have been addressed

Reviewer #2: All comments have been addressed

2. Is the manuscript technically sound, and do the data support the conclusions?

Reviewer #1: Yes

Reviewer #2: Yes

3. Has the statistical analysis been performed appropriately and rigorously? 

Reviewer #1: Yes

Reviewer #2: N/A

4. Have the authors made all data underlying the findings in their manuscript fully available?

Reviewer #1: Yes

Reviewer #2: Yes

5. Is the manuscript presented in an intelligible fashion and written in standard English?

Reviewer #1: Yes

Reviewer #2: Yes

6. Review Comments to the Author

Reviewer #1: I have no further comments. The authors did all revisors smoothly with excellent answers and I feel the paper can be accepted in tis current form.

Reviewer #2: The authors had addressed all the concerns raised by the reviewers in the earlier review process. So, from my side there is no issue in getting this work published in this journal.

7. PLOS authors have the option to publish the peer review history of their article (what does this mean?). If published, this will include your full peer review and any attached files.

Reviewer #1: No

Reviewer #2: **Yes: **Bikash Ranjan Behera

---

## [Editor Report · Acceptance letter]

29 Jan 2024

PONE-D-23-33755R1 

PLOS ONE

Dear Dr. Khan, 

I'm pleased to inform you that your manuscript has been deemed suitable for publication in PLOS ONE. Congratulations! Your manuscript is now being handed over to our production team.

Kind regards, 

on behalf of

Dr. Yuan-Fong Chou Chau 

Academic Editor

PLOS ONE